# Active Structure Learning of Causal DAGs via Directed Clique Trees

**Chandler Squires**
LIDS, MIT
MIT-IBM Watson AI Lab
csquires@mit.edu

**Sara Magliacane**
MIT-IBM Watson AI Lab
IBM Research
sara.magliacane@gmail.com

**Kristjan Greenewald**
MIT-IBM Watson AI Lab
IBM Research
kristjan.h.greenewald@ibm.com

**Dmitriy Katz**
MIT-IBM Watson AI Lab
IBM Research
dkatzrog@us.ibm.com

**Murat Kocaoglu**
MIT-IBM Watson AI Lab
IBM Research
murat@ibm.com

**Karthikeyan Shanmugam**
MIT-IBM Watson AI Lab
IBM Research
karthikeyan.shanmugam2@ibm.com

## Abstract

A growing body of work has begun to study intervention design for efficient structure learning of causal directed acyclic graphs (DAGs). A typical setting is a *causally sufficient* setting, i.e. a system with no latent confounders, selection bias, or feedback, when the essential graph of the observational equivalence class (EC) is given as an input and interventions are assumed to be noiseless. Most existing works focus on *worst-case* or *average-case* lower bounds for the number of interventions required to orient a DAG. These worst-case lower bounds only establish that the largest clique in the essential graph *could* make it difficult to learn the true DAG. In this work, we develop a *universal* lower bound for single-node interventions that establishes that the largest clique is *always* a fundamental impediment to structure learning. Specifically, we present a decomposition of a DAG into independently orientable components through *directed clique trees* and use it to prove that the number of single-node interventions necessary to orient any DAG in an EC is at least the sum of half the size of the largest cliques in each chain component of the essential graph. Moreover, we present a two-phase intervention design algorithm that, under certain conditions on the chordal skeleton, matches the optimal number of interventions up to a multiplicative logarithmic factor in the number of maximal cliques. We show via synthetic experiments that our algorithm can scale to much larger graphs than most of the related work and achieves better worst-case performance than other scalable approaches. [1]

## 1  Introduction

Causal modeling is an important tool in medicine, biology and econometrics, allowing practitioners to predict the effect of actions on a system and the behavior of a system if its causal mechanisms change due to external factors (Pearl, 2009; Spirtes et al., 2000; Peters et al., 2017). A commonly-used model

is the directed acyclic graph (DAG), which is capable of modeling *causally sufficient* systems, i.e. systems with no latent confounders, selection bias, or feedback. However, even in this favorable setup, a causal model cannot (in general) be fully identified from observational data alone; in these cases experimental ("interventional") data is necessary to resolve ambiguities about causal relationships.

In many real-world applications, interventions may be time-consuming or expensive, e.g. randomized controlled trials or gene knockout experiments. These settings crucially rely on *intervention design*, i.e. finding a cost-optimal set of interventions that can fully identify a causal model. Recently, many methods have been developed for intervention design under different assumptions (He & Geng, 2008; Hyttinen et al., 2013; Shanmugam et al., 2015; Kocaoglu et al., 2017; Lindgren et al., 2018).

In this work we extend the Central Node algorithm of Greenewald et al. (2019) to learn the structure of causal graphs in a *causally sufficient* setting from interventions on single variables for both noiseless and noisy interventions. Noiseless interventions are able to deterministically orient a set of edges, while noisy interventions result in a posterior update over a set of compatible graphs. We also focus only on interventions with a single target variable, i.e. *single-node interventions*, but as opposed to (Greenewald et al., 2019) which focuses on limited types of graphs, we allow for general DAGs but only consider noiseless interventions. In particular, we focus on *adaptive* intervention design, also known as *sequential* or *active* (He & Geng, 2008), where the result of each intervention is incorporated into the decision-making process for later interventions. This contrasts with *passive* intervention design, for which all interventions are decided beforehand.

**Universal lower bound.** Our key contribution is to show that the problem of fully orienting a DAG with single-node interventions is equivalent to fully orienting special induced subgraphs of the DAG, called *residuals* (Theorem 1 below). Given this decomposition, we prove a universal lower bound on the minimum number of single-node interventions necessary to fully orient *any* DAG in a given Markov Equivalence Class (MEC), the set of graphs that fit the observational distribution. This lower bound is equal to the sum of half the size of the largest cliques in each chain component of the essential graph (Theorem 2). This result has a surprising consequence: the largest clique is *always* a fundamental impediment to structure learning. In comparison, prior work (Hauser & Bühlmann, 2014; Shanmugam et al., 2015) established *worst-case* lower bounds based on the maximum clique size, which only implied that the largest clique in each chain component of the essential graph *could* make it difficult to learn the true DAG.

**Intervention policy.** We also propose a novel two-phase single-node intervention policy. The first phase, based on the Central Node algorithm, uses properties of *directed clique trees* (Definition 2) to reduce the identification problem to identification within the (DAG dependent) residuals. The second phase then completes the orientations within each residual. We cover the condition of *intersection-incomparability* for the chordal skeleton of a DAG (Kumar & Madhavan (2002) introduce this condition in the context of graph theory) . We show that under this condition, our policy uses at most $O(\log \mathcal{C}_{\max})$ times as many interventions as are used by the (DAG dependent) optimal intervention set, where $\mathcal{C}_{\max}$ is the greatest number of maximal cliques in any chain component (Theorem 3).

Finally, we evaluate our policy on general synthetic DAGs. We find that our intervention policy performs comparably to intervention policies in previous work, while being much more scalable than most policies and adapting more effectively to the difficulty of the underlying identification problem.

## 2   Preliminaries

We briefly review our notation and terminology for graphs. A mixed graph $G$ is a tuple of vertices $V(G)$, directed edges $D(G)$, bidirected edges $B(G)$, and undirected edges $U(G)$. Directed, bidirected, and undirected edges between vertices $i$ and $j$ in $G$ are denoted $i \to_G j$, $i \leftrightarrow_G j$, and $i -_G j$, respectively. We use asterisks as wildcards for edge endpoints, e.g., $i * \to_G j$ denotes either $i \to_G j$ or $i \leftrightarrow_G j$. A *directed cycle* in a mixed graph is a sequence of edges $i * \to_G \dots * \to_G i$ with at least one directed edge. A mixed graph is a *chain graph* if it has no directed cycles and $B(G) = \emptyset$, and a chain graph is called a *directed acyclic graph* (*DAG*) if we also have $U(G) = \emptyset$. An *undirected graph* is a mixed graph with $B(G) = \emptyset$ and $D(G) = \emptyset$.

**DAGs and ($\mathcal{I}$-)Markov equivalence.** DAGs are used to represent causal models (Pearl, 2009). Each vertex $i$ is associated with a random variable $X_i$. The *skeleton* of graph $D$, $\mathrm{skel}(D)$, is the undirected graph with the same vertices and adjacencies as $D$. A distribution $f$ is *Markov* w.r.t. a DAG $D$ if it

factors as $f(X) = \prod_{i \in V(D)} f(X_i \mid X_{\text{pa}_D(i)})$. Two DAGs $D_1$ and $D_2$ are called *Markov equivalent* if all positive distributions $f$ which are Markov to $D_1$ are also Markov to $D_2$ and vice versa. The set of DAGs that are Markov equivalent to $D$ is the *Markov equivalence class* (MEC), denoted as $[D]$. $[D]$ is represented by a chain graph called the *essential graph* $\mathcal{E}(D)$, which has the same skeleton as $D$, with directed edges $i \rightarrow_{\mathcal{E}(D)} j$ if $i \rightarrow_{D'} j$ for all $D' \in [D]$, and undirected edges otherwise. Given an intervention $I \subseteq V(D)$, the distributions $(f^{\text{obs}}, f^I)$ are *I-Markov* to $D$ if $f^{\text{obs}}$ is Markov to $D$ and $f^I$ factors as

$$f^I(X) = \prod_{i \notin I} f^{\text{obs}}(X_i \mid X_{\text{pa}_D(i)}) \prod_{i \in I} f^I(X_i \mid X_{\text{pa}_D(i)})$$

where $\text{pa}_D(i)$ represents the set of parents of vertex $i$ in the DAG $D$. Given a list of interventions $\mathcal{I} = [I_1, \dots, I_M]$, the set of distributions $\{f^{\text{obs}}, f^{I_1}, \dots, f^{I_M}\}$ is *$\mathcal{I}$-Markov* to a DAG $D$ if $(f^{\text{obs}}, f^{I_m})$ is $I_m$-Markov to $D$ for $\forall m = 1 \dots M$. The *$\mathcal{I}$-Markov equivalence class* of $D$ ($\mathcal{I}$-MEC), denoted as $[D]_{\mathcal{I}}$, can be represented by the *$\mathcal{I}$-essential graph* $\mathcal{E}_{\mathcal{I}}(D)$ with the same adjacencies as $D$ and $i \rightarrow_{\mathcal{E}_{\mathcal{I}}(D)} j$ if $i \rightarrow_{D'} j$ for all $D' \in [D]_{\mathcal{I}}$.

The edges which are *undirected* in the essential graph $\mathcal{E}(D)$, but *directed* in the $\mathcal{I}$-essential graph $\mathcal{E}_{\mathcal{I}}(D)$, are the edges which are learned from performing the interventions in $\mathcal{I}$. In the special case of a single-node intervention, the edges learned are all of those *incident* to the intervened node, along with any edges learned via the set of logical constraints known as Meek rules Appendix A.

**Structure of essential graphs.** We now report a known result that proves that any intervention policy can split essential graphs in components that can be oriented independently. The *chain components* of a chain graph $G$, denoted $\text{CC}(G)$, are the connected components of the graph after removing its directed edges. These chain components are then clearly undirected graphs. An undirected graph is *chordal* if every cycle of length greater than 3 has a *chord*, i.e., an edge between two non-consecutive vertices in the cycle.

**Lemma 1** (Hauser & Bühlmann (2014)). *Every $\mathcal{I}$-essential graph is a chain graph with chordal chain components. Orientations in one chain component do not affect orientations in other components.*

**Definition 1.** *A DAG whose essential graph has a single chain component is called a moral DAG.*

In many of the following results we will consider moral DAGs, since once we can orient moral DAGs we can easily generalize to general DAGs through these results.

**Intervention Policies.** An *intervention policy* $\pi$ is a (possibly randomized) map from ($\mathcal{I}$-)essential graphs to interventions. An intervention policy is *adaptive* if each intervention $I_m$ is decided based on information gained from previous interventions, and *passive* if the whole set of interventions $\mathcal{I}$ is decided prior to any interventions being performed. An intervention is *noiseless* if the intervention set $\mathcal{I}$ collapses the set of compatible graphs exactly to the $\mathcal{I}$-MEC, while *noisy* interventions simply induce a posterior update on the distribution over compatible graphs. Most policies assume that the MEC is known (e.g., it has been estimated from observational data) and interventions are noiseless; this is true of our policy too. Moreover, we focus only on interventions on a single target variable, i.e. *single-node interventions*. We discuss previous work on intervention policies in Section 6.

## 3  Universal lower-bound in the number of single-node interventions

In this section we prove a lower-bound on any possible single-node policy (Theorem 2) by decomposing the complete orientation of a DAG in terms of the complete orientation of smaller independent subgraphs, called *residuals* (Theorem 1), defined on a novel graphical structure, *directed clique trees* (DCTs). We provide all proofs in the Appendix.

First, we review the standard definitions of clique trees and clique graphs for undirected chordal graphs (see also (Galinier et al., 1995)). A *clique* $C \subseteq V(G)$ is a subset of the nodes with an edge between each pair of nodes. A clique $C$ is *maximal* if $C \cup \{v\}$ is not a clique for any $v \in V(G) \setminus C$. The set of maximal cliques of $G$ is denoted $\mathcal{C}(G)$. The *clique number* of $G$ is $\omega(G) = \max_{C \in \mathcal{C}} |C|$. A *clique tree* (aka a junction tree) $T_G$ of a chordal graph is a tree with vertices $\mathcal{C}(G)$ that satisfies the *induced subtree property*, i.e., for any $v \in V(G)$, the induced subgraph on the set of cliques containing $v$ is a tree. A chordal graph can have multiple clique trees, so we denote the set of all clique trees of $G$ as $\mathcal{T}(G)$. A *clique graph* $\Gamma_G$ is the graph union of all clique trees, i.e. the undirected graph

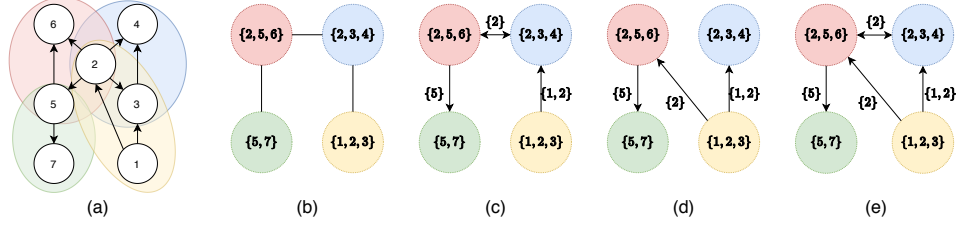

Figure 1: A moral DAG (a), one of its clique trees (b), its two DCTs (c-d) and the DCG (e).

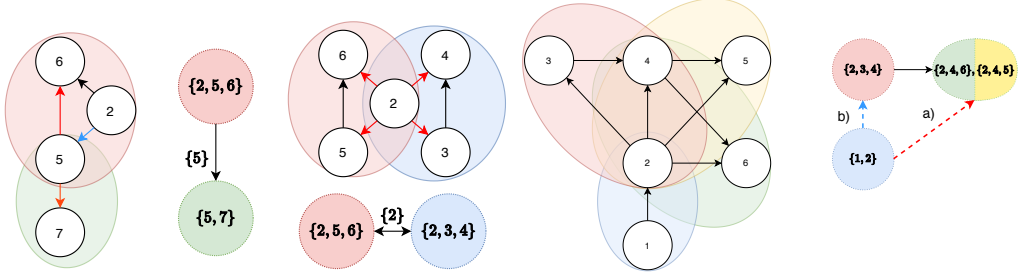

Figure 2: Examples of edge orientations.

Figure 3: A DAG and its CDCTs with (using only edge a) and without arrow-meets (edge b).

with $V(\Gamma_G) = \mathcal{C}(G)$ and $U(\Gamma_G) = \cup_{T \in \mathcal{T}(G)} U(T)$. A useful characterization of the clique trees of $G$ are as the max-weight spanning trees of the weighted clique graph $W_G$ (Koller & Friedman, 2009), which is a complete graph over vertices $\mathcal{C}(G)$, with the edge $C_1 -_{W_G} C_2$ having weight $|C_1 \cap C_2|$.

Given a moral DAG $D$, we can trivially define its clique trees $\mathcal{T}(D)$ as the clique trees of its skeleton $G = \text{skel}(D)$, i.e. $\mathcal{T}(G)$. For example, in Fig. 1 (a) we show a DAG, where we have chosen a color for each of the cliques, while in Fig. 1 (b) we show one of its clique trees. We now define a directed counterpart to clique trees based on the orientations in the underlying DAG:

**Definition 2.** *A* directed clique tree $T_D$ *of a moral DAG $D$ has the same vertices and adjacencies as a clique tree $T_G$ of $G = \text{skel}(D)$. For each ordered pair of adjacent cliques $C_1 *\!-\!* C_2$ we orient the edge mark of $C_2$ as:*

- $C_1 *\!\to C_2$, *if* $\forall v_{12} \in C_1 \cap C_2$ *and* $\forall v_2 \in C_2 \setminus C_1$, *we have* $v_{12} \to_D v_2$ *in the DAG $D$;*
- $C_1 *\!- C_2$ *otherwise, i.e. if there exists at least one incoming edge from $C_2 \setminus C_1$ into $C_1 \cap C_2$,*

*where we recall that $*$ denotes a wildcard for an edge. Thus, the above conditions only decide the presence or absence of an arrowhead at $C_2$; the presence or absence of an arrowhead at $C_1$ is decided when considering the reversed order.*

A DAG can have multiple directed clique trees (DCTs), as shown in Fig. 1 (c) and (d). In figures, we annotate edges with the intersection between cliques. Fig. 1 (c) represents the directed clique tree corresponding to the standard clique tree in Fig. 1 (b). In Fig. 2 we show in detail the orientations for two of the directed clique edges following Definition 2, the red edges are outcoming from the clique intersection, while the blue edge is incoming in the intersection. Definition 2 also implies each edge that is shared between two different clique trees has a unique orientation (since it is based on the underlying DAG), so we can define the directed clique graph (DCG) $\Gamma_D$ of a moral DAG $D$ as the graph union of all directed clique trees of $D$. We show an example of a DCG in Fig. 1(e). As can be seen in the examples in Fig. 1, DCTs can contain directed and bidirected edges, and, as we prove in Appendix C, no undirected edgees. We define the bidirected components of a DCT as:

**Definition 3.** *The* bidirected components *of $T_D$, $\mathcal{B}(T_D)$, are the connected components of $T_D$ after removing directed edges.*

Another structure that can happen in a DCT is when two arrows meet at the same clique. To avoid confusing associations with colliders in DAGs, we call these structures in DCTs *arrow-meets*. Arrow-meets will prove to be challenging for our algorithms, so we introduce *intersection incomparability* and prove that in case it holds there can be no arrow-meets:

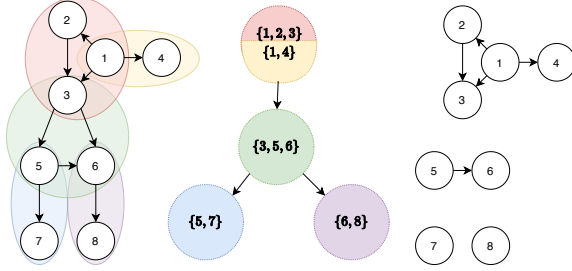

Figure 4: A DAG, its CDCT and its residuals.

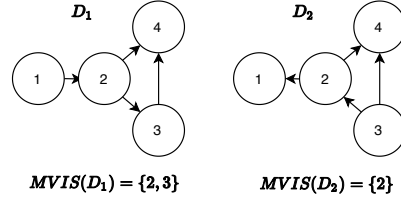

Figure 5: DAGs in the same MEC with $m(D_1) \neq m(D_2)$.

**Definition 4.** *A pair of edges $C_1 -_{T_G} C_2$ and $C_2 -_{T_G} C_3$ are* intersection comparable *if $C_1 \cap C_2 \subseteq C_2 \cap C_3$ or $C_1 \cap C_2 \supseteq C_2 \cap C_3$. Otherwise they are* intersection incomparable.

For example, in Fig. 1 (e), the edges $\{2,5,6\}\leftrightarrow\{2,3,4\}$ and $\{2,3,4\}\leftarrow\{1,2,3\}$ are intersection comparable, since $\{2\} \subset \{1,2\}$, while $\{2,5,6\}\leftrightarrow\{2,3,4\}$ and $\{2,5,6\}\rightarrow\{5,7\}$ are intersection incomparable, since $\{2\} \not\subseteq \{5\}$ and $\{5\} \not\subseteq \{2\}$.

**Proposition 1.** *Suppose $C_1 *\!\!\rightarrow_{T_D} C_2$ and $C_2 \leftarrow\!* _{T_D} C_3$ in $T_D$. Then these edges are intersection comparable. Equivalently in the contrapositve, if $C_1 *\!\!\rightarrow_{T_D} C_2$ and $C_2 *\!\!-\!* _{T_D} C_4$ are intersection incomparable, we can immediately deduce that $C_2 \rightarrow_{T_D} C_4$.*

Bidirected components do not have a clear ordering, so we contract them into single nodes in a contracted DCT (CDCT), and prove we can always construct a tree-like CDCT for any moral DAG:

**Definition 5.** *The* contracted directed clique tree (CDCT) *$\tilde{T}_D$ of a DCT $T_D$ is a graph on the vertex set $B_1, B_2 \ldots B_K \in \mathcal{B}(T_D)$ with $B_1 \rightarrow_{\tilde{T}_D} B_2$ if $C_1 \rightarrow_{T_D} C_2$ for any clique $C_1 \in B_1$ and $C_2 \in B_2$.*

**Lemma 2.** *For any moral DAG $D$, one can always construct a CDCT with no arrow-meets.*

In particular, one can adapt Kruskal's algorithm for finding a max-weight spanning tree to construct a DCT from the weighted clique graph and then contract it, as shown in detail in Algorithm 3 in Appendix D. In Fig. 3 we show an example of a CDCT with arrow-meets (represented by the black edge and the edge labelled "a") and its equivalent no arrow-meets version (represented by the black edge and the edge "b") . Since we can always construct a CDCT with no arrow-meets, we assume w.l.o.g. that the CDCT is a tree. The CDCT allows us to define a decomposition of a moral DAG into independently orientable components. We call these components *residuals*, since they extend the notion of residuals in rooted, undirected clique trees (Vandenberghe et al., 2015). Formally:

**Definition 6.** *For a tree-like CDCT $\tilde{T}_D$ of a moral DAG $D$, the* residual *of its node $B$ is defined as $\mathrm{Res}_{\tilde{T}_D}(B) = D|_{B\backslash P}$, where $P$ is its parent in $\tilde{T}_D$ (or if there is none, $P = \emptyset$) and $D|_{B\backslash P}$ is the induced subgraph of $D$ over the subset of $V(D)$ that are assigned to $B$ but not to $P$. We denote the set of all residuals of $\tilde{T}_D$ by $\mathcal{R}(\tilde{T}_D)$.*

Intuitively this describes the subgraphs in which we cut all edges that are captured in the CDCT, as shown in Fig. 4. We now generalize our results from a moral DAG to a general DAG. Surprisingly, we show that orienting all of the residuals for all chain components in the essential graph is both necessary and sufficient to *completely orient any DAG*. We start by introducing a *VIS*:

**Definition 7.** *Given a general DAG $D$, a* verifying intervention set (VIS) *is a set of single-node interventions $\mathcal{I}$ that fully orients the DAG starting from an essential graph, i.e. $\mathcal{E}_{\mathcal{I}}(D) = D$. A minimal VIS (MVIS) is a VIS of minimal size. We denote the size of the minimal VIS for $D$ as $m(D)$.*

For each DAG there are many possible VISes. A trivial VIS for any DAG is just the set of all of its nodes. In general, we are more interested in MVISes, which are also not necessarily unique for a DAG. For example, the DAG in Fig. 4 has four MVISes: $\{1,3,5\}$, $\{1,3,6\}$, $\{2,4,5\}$, and $\{2,4,6\}$.

We now show that finding a VIS for any DAG $D$ can be decomposed twice: first we can create a separate task of finding a VIS for each of the chain components $G$ of its essential graph $\mathcal{E}(D)$, and then for each $G$ we can create a tree-like CDCT and find independently a VIS for each of its residuals:

**Theorem 1.** *A single-node intervention set is a VIS for any general DAG $D$ iff it contains VISes for each residual $R \in \mathcal{R}(\tilde{T}_G)$ for all chain components $G \in \mathcal{CC}(\mathcal{E}(D))$ of its essential graph $\mathcal{E}(D)$.*

An MVIS of $D$ will then contain only the MVISes of each residual of each chain component. An algorithm using this decomposition to compute an MVIS is given in Appendix F. In general, the size of an MVIS of $D$ cannot be calculated from just its essential graph, as shown by the two graphs in Fig. 5. Instead, we propose a *universal lower bound* that holds for all DAGs in the same MEC:

**Theorem 2.** *Let $D$ be any DAG. Then $m(D) \geq \sum_{G \in cc(\mathcal{E}(D))} \left\lfloor \frac{\omega(G)}{2} \right\rfloor$, where $\omega(G)$ is the size of the largest clique in each of the chain components $G$ of the essential graph $\mathcal{E}(D)$.*

We reiterate how this bound is different from previous work. For a fixed MEC $[D]$ with essential graph $\mathcal{E}$, it is easy to construct $D^* \in [D]$ such that $m(D^*) \geq \sum_{G \in cc(\mathcal{E}(D))} \left\lfloor \frac{\omega(G)}{2} \right\rfloor$ by picking the largest clique in each chain component to be the upstream-most clique. The bound in Theorem 2 gives a much stronger result: *any* choice of DAG in the MEC requires this many single-node interventions.

## 4  A two-phase intervention policy based on DCTs

While in the previous section we started from a known DAG $D$ to construct a CDCT and then proved an universal lower bound on $m(D)$, in this section we focus on intervention design to learn the orientations of an unknown DAG starting from its observational essential graph. Theorem 1 proves that to orient a DAG $D$, we only need to orient the residuals for each of its essential graph chain components. The definition of residuals requires the knowledge of a tree-like CDCT for each component, which can be easily derived from the directed clique graph (DCG) (e.g. through Algorithm 3 in Appendix D). So, we propose a two phase policy, in which the first phase uses interventions to identify the DCG of each chain components, while the second phase uses interventions to orient each of the residuals, as described in Algorithm 1. We now focus on describing the first phase of the algorithm and start by introducing two types of abstract, higher-level interventions.

**Definition 8.** *A* clique-intervention *on a clique $C$ is a series of single-node interventions that suffices to learn the orientation of all edges in $\Gamma_D$ that are incident on $C$. An* edge-intervention *on an edge $C_1 -_{T_G} C_2$ is a series of single-node interventions that suffices to learn the orientation of $C_1 -_{T_D} C_2$.*

A trivial clique-intervention is intervening on all of $C$, and a trivial edge-intervention is intervening on all of $C_1 \cap C_2$. The clique- and edge- interventions we use in practice are outlined in Appendix H.

| **Algorithm 1** DCT POLICY | **Algorithm 2** FINDDCG |
|---|---|
| 1: **Input:** essential graph $\mathcal{E}(D)$ | 1: **Input:** clique graph $\Gamma_G$ |
| 2: **for** component $G$ in $CC(\mathcal{E}(D))$ **do** | 2: let $\Gamma_D = \Gamma_G$ |
| 3:   create clique graph $\Gamma_G$ | 3: **while** $\Gamma_D$ has undirected edges **do** |
| 4:   $\Gamma_D = $ FINDDCG$(\Gamma_G)$ | 4:   let $T$ be a maximum-weight spanning tree of the undirected component of $\Gamma_D$ |
| 5:   convert $\Gamma_D$ to a CDCT $\tilde{T}_D$ | 5:   let $C$ be a central node of $T$ |
| 6:   **for** clique $C$ in $\tilde{T}_D$ **do** | 6:   perform a clique-intervention on $C$ |
| 7:     Let $R = \text{Res}_{\tilde{T}_D}(C)$ | 7:   let $P_{\text{up}}(C) = $ IDENTIFYUPSTREAM$(C)$ |
| 8:     Intervene on nodes in $V(R)$ until $R$ is fully oriented | 8:   let $S = V(B_T^{C:P_{\text{up}}(C)})$ |
| 9:   **end for** | 9:   **while** $\Gamma_D$ has unoriented incident to $\mathcal{C} \setminus S$ **do** |
| 10: **end for** | 10:     propagate edges in $\Gamma_D$ |
| 11: **return** completely oriented $D$ | 11:     perform an edge-intervention on an edge $C_1 -_{\Gamma_G} C_2$ with $C_1 \in \mathcal{C} \setminus S$ |
| | 12:   **end while** |
| | 13: **end while** |
| | 14: **return** $\Gamma_D$ |

The first phase of our algorithm, described in Algorithm 2, is inspired by the Central Node algorithm (Greenewald et al., 2019). This algorithm operates over a tree, so we will have to use a spanning tree:

**Definition 9.** *(Greenewald et al., 2019) Given a tree $T$ and a node $v \in V(T)$, we divide $T$ into branches w.r.t. $v$. For a node $w$ adjacent to $v$, the branch $B_T^{(v:w)}$ is the connected component of $T - \{v\}$ that contains $w$. A* central node $c$ *is a node for which $\forall w$ adjacent to $c : |B_T^{(c:w)}| \leq |\frac{V(T)}{2}|$.*

While our algorithm works for general graphs, it will help our intuition to first assume that $\Gamma_G$ is intersection-incomparable. In this case, there are no arrow-meets in $\Gamma_D$ by Prop. 1, nor in any of the directed clique trees. Thus, after each clique-intervention on a central node $C$, there will be only one parent clique upstream and the algorithm will orient at least half of the remaining unoriented edges by repeated application of Prop. 1. For the intersection-comparable case, two steps can go wrong. First, after a clique-intervention on $C$, we may find that $C$ has multiple parents in $\Gamma_D$ (i.e. $C$ is at an arrows-meet). We can prove that even in this case, there is always a single "upstream" branch, identified via the `IdentifyUpstream` procedure, described in Appendix I, which performs edge-interventions on a subset of the parents. A second step which may go wrong is in the propagation of orientations along the downstream branches, which halts when encountering intersection-incomparable edges. In this case, we simply kickstart further propagation by performing an edge-intervention.

The size of the problem is cut in half after each clique-intervention, so that we use at most $\sum_{G \in \text{cc}(\mathcal{E}(D))} \lceil \log_2(|\mathcal{C}(G)|) \rceil$ clique-interventions, where $\mathcal{C}(G)$ is the set of maximal cliques for $G$. Furthermore, if $\Gamma_G$ is intersection-incomparable we use no edge-interventions (see Lemma 8 in Appendix J). The second phase of the algorithm then orients the residuals and uses at most $\sum_{G \in \text{cc}(\mathcal{E}(D))} \sum_{C \in \mathcal{C}(G)} |\text{Res}_{\tilde{T}_G}(C)| - 1$ single-node interventions (see Lemma 9 in Appendix J).

**Theorem 3.** *Assuming $\Gamma_G$ is intersection-incomparable, Algorithm 1 uses at most $(3\lceil \log_2 \mathcal{C}_{max} \rceil + 2)m(D)$ single-node interventions, where $\mathcal{C}_{max} = \max_{G \in \text{cc}(\mathcal{E}(D))} |\mathcal{C}(G)|$.*

In the extreme case in which the essential graph is a tree, a single intervention on the root node can orient the tree, so $m(D) = 1$, and $|\mathcal{C}| = |V(D)| - 1$, so Theorem 3 says that Algorithm 1 uses $O(\log(p))$ interventions, which is the scaling of the Bayes-optimal policy for the uniform prior as discussed in Greenewald et al. (2019).

**Remark on intersection-incomparability.** Intersection-incomparable chordal graphs were introduced as "uniquely representable chordal graphs" in Kumar & Madhavan (2002). This class was shown to include familiar classes of graphs such as proper interval graphs. While the assumption of intersection-incomparability is necessary for our analysis of the DCT policy, the policy still performs well on intersection-comparable graphs as demonstrated in Section 5. This suggests that the restriction may be an artifact of our analysis, and the result of Theorem 3 may hold more generally.

## 5 Experimental Results

We evaluate our policy on synthetic graphs of varying size. To evaluate the performance of a policy on a specific DAG $D$, relative to $m(D)$, the size of its smallest VIS (MVIS), we adapt the notion of *competitive ratio* from online algorithms (Borodin et al., 1992; Daniely & Mansour, 2019). We use $\iota_D(\pi)$ to denote the expected size of the VIS found by policy $\pi$ for the DAG $D$, and define our evaluation metric as:

**Definition 10.** *The* instance-wise competitive ratio *(ic-ratio) of an intervention policy $\pi$ on $D$ is* $\text{R}(\pi, D) = \frac{\iota_{D'}(\pi)}{m(D')}$. *The* competitive ratio *on an MEC $[D]$ is* $\text{R}(\pi) = max_{D' \in [D]} \frac{\iota_{D'}(\pi)}{m(D')}$.

The instance-wise competitive ratio of a policy on a DAG $D$ simply measures the number of interventions used by the policy *relative* to the number of interventions used by the best policy *for that DAG*, i.e., the policy which guesses that $D$ is the true DAG and uses exactly a MVIS of $D$ to verify this guess. Thus, a *lower* ic-ratio is better, and an ic-ratio of 1 is the best possible. In order to compute the ic-ratio on $D$, we must compute $m(D)$, the size of a MVIS for $D$. In our experiments, we use our DCT characterization of VIS's from Theorem 1 to decompose the DAG into its residuals, each of whose MVIS's can be computed efficiently. We describe this procedure in Appendix F.

**Smaller graphs.** For our evaluation on smaller graphs, we generate random connected moral DAGs using the following procedure, which is a modification of Erdös-Rényi sampling that guarantees that the graph is connected. We first generate a random ordering $\sigma$ over vertices. Then, for the $n$-th node in the order, we set its indegree to be $X_n = \max(1, \text{Bin}(n - 1, \rho))$, and sample $X_n$ parents uniformly from the nodes earlier in the ordering. Finally, we chordalize the graph by running the elimination algorithm (Koller & Friedman, 2009) with elimination ordering equal to the reverse of $\sigma$.

We compare the `OptSingle` policy (Hauser & Bühlmann, 2014), the Minmax and Entropy strategy of He & Geng (2008), called `MinmaxMEC` and `MinmaxEntropy`, respectively, and the coloring-based strategy of Shanmugam et al. (2015), called `Coloring`. We also introduce a baseline that picks

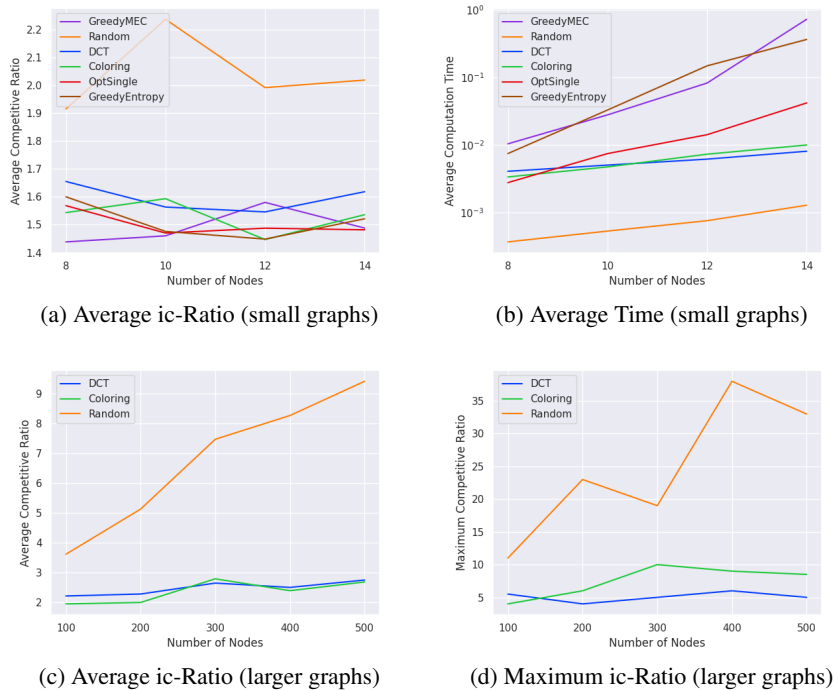

(a) Average ic-Ratio (small graphs)

(b) Average Time (small graphs)

(c) Average ic-Ratio (larger graphs)

(d) Maximum ic-Ratio (larger graphs)

Figure 6: Comparison of intervention policies over 100 synthetic DAGs.

randomly among non-dominated[2] nodes in the $\mathcal{I}$-essential graph, called the *non-dominated random* (ND-Random) strategy. As the name suggests, dominated nodes are easily proven to be non-optimal interventions, so ND-Random is a more fair baseline than simply picking randomly amongst nodes.

In Fig. 6a and Fig. 6b, we show the average ic-ratio and the average run-time for each of the algorithms. In terms of average ic-ratio, all algorithms aside from ND-Random perform comparably, using on average 1.4-1.7x more interventions than the smallest MVIS. However, the computation time grows quite quickly for GreedyMEC, GreedyEntropy, and OptSingle. This is because, when scoring a node as a potential intervention target, each of these algorithms iterates over all possible parent sets of the node. Moreover, the GreedyMEC and GreedyEntropy policies then compute the sizes of the resulting interventional MECs, which can grow superexponentially in the number of nodes (Gillispie & Perlman, 2013). In Appendix K, we show that in the same setting, OptSingle takes >10 seconds per graph for just 25 nodes, whereas Coloring, DCT, and Random remain under .1 seconds per graph.

**Larger graphs.** For our evaluation on large tree-like graphs, we create random moral DAGs of $n = 100, \ldots, 300$ nodes using the following procedure. We generate a complete directed 4-ary tree on $n$ nodes. Then, we sample an integer $R \sim U(2, 5)$ and add $R$ edges to the tree. Finally, we find a topological order of the graph by DFS and triangulate the graph using that order. This ensures that the graph retains a nearly tree-like structure, making $m(D)$ small compared to the overall number of nodes. In Fig. 6c and Fig. 6d, we show the average and maximum competitive ratio (computation time is given in Appendix K). For the average graph, our DCT policy and the Coloring policy use only 2-3 times as many interventions as the theoretical lower bound. Moreover, the worst competitive ratio experienced by the DCT algorithm is significantly smaller than the worst ratio experienced by the Coloring policy, which suggests that our policy is more adaptive to the underlying difficulty of the identification problem.

# 6   Related Work

Intervention policies fall under two distinct, but related goals. The first is: given a fixed number of interventions, learn as much as possible about the underlying DAG. This goal is explored in Ghassami et al. (2017, 2018) and Hauser & Bühlmann (2014). The second goal, which is the one considered in this paper, is *minimum-cost identification*: completely learn the underlying DAG using the least number of interventions. We review previous work on policies operating under this objective. As before, we use $\iota_D(\pi)$ to denote the expected size of the VIS found by policy $\pi$.

We define $\Pi_K$ as the set of policies using interventions with at most $K$ target variables, i.e., $|I_m| \leq K$ for $I_m \in \mathcal{I}$. We use $\Pi_\infty$ to represent policies allowing for interventions of unbounded size. A policy $\pi$ is *K-node minimax optimal* for an MEC $[D]$ if $\pi \in \arg\min_{\pi' \in \Pi_K} \max_{D' \in [D]} \iota_{D'}(\pi')$. Informally, this is the policy $\pi$ that in the worst-case scenario (the DAG in the MEC that requires the most interventions under $\pi$) ends up requiring the least interventions. A policy is *K-node Bayes-optimal* for an MEC $[D]$ and a prior $\mathbb{P}_\mathsf{D}$ supported only on the MEC $[D]$ if $\pi \in \arg\min_{\pi' \in \Pi_K} \mathbb{E}_{\mathbb{P}_\mathsf{D}}[\iota_\mathsf{D}(\pi')]$

In the special cases of $K = 1$ and $K = \infty$, we replace $K$-node by *single-node* and *unbounded*, respectively. Much recent work explores intervention policies under a variety of objectives and constraints. Eberhardt (2007) introduced passive, minimax-optimal intervention policies for single-node, $K$-node, and unbounded interventions in both the causally sufficient and causally insufficient case, when the MEC is not known. They also give a passive, unbounded intervention policy when the MEC *is* known, and conjectures a minimax lower bound of $\lceil \frac{\omega(\mathcal{E}(D))}{2} \rceil$ on $\iota_D(\pi)$ for such policies. Hauser & Bühlmann (2014) prove this bound by developing a passive, unbounded minimax-optimal policy. Shanmugam et al. (2015) develop a $K$-node minimax lower bound of $\frac{\omega(\mathcal{E}(D))}{K} \log_{\frac{\omega(\mathcal{E}(D))}{K}} \omega(\mathcal{E}(D))$ based on separating systems. Kocaoglu et al. (2017) develop a passive, unbounded minimax-optimal policy when interventions have distinct costs (where $\iota_D(\pi)$ is replaced by the total cost of all interventions.) Greenewald et al. (2019) develop an adaptive $K$-node intervention policy for noisy interventions which is within a small constant factor of the Bayes-optimal intervention policy, but the policy is limited to the case in which the chain components of the essential graph are trees. It is important to note that all of these previous works give *minimax* optimal policies, i.e. they focus on minimizing the interventions used in the *worst* case over the MEC. In contrast, our result in Theorem 3 is *competitive*, holding for *every* DAG in the MEC, and shows that the largest clique is still a fundamental impediment to structure learning. However, the current result holds only in the single-node case, whereas previous work allows for larger interventions.

Finally, we note an interesting conceptual connection to Ghassami et al. (2019), which uses undirected clique trees as a tool for counting and sampling from MECs, suggesting that clique trees and their variants, such as DCTs, may be broadly useful for a variety of DAG-related tasks.

# 7   Discussion

We presented a decomposition of a moral DAG into residuals, each of which must be oriented independently of one another. We use this decomposition to prove that for any DAG $D$ in a MEC with essential graph $\mathcal{E}$, at least $\sum_{G \in \mathtt{CC}(\mathcal{E})} \lfloor \frac{\omega(G)}{2} \rfloor$ interventions are necessary to orient $D$, where $\mathtt{CC}(\mathcal{E})$ denotes the chain components of $\mathcal{E}$ and $\omega(G)$ denotes the clique number of $G$. We introduced a novel two-phase intervention policy, which first uses a variant of the Central-Node algorithm to obtain orientations for the directed clique graph $\Gamma_D$, then orients within each residual. We showed that under certain conditions on the chain components of $\mathcal{E}$, this intervention policy uses at most $(3 \log_2 \mathcal{C}_{\max} + 2)$ times as many interventions as the optimal intervention set. Finally, we showed on synthetic graphs that our intervention policy is more scalable than most existing policies, with comparable performance to the coloring-based policy of Shanmugam et al. (2015) in terms of average ic-ratio and better performance in terms of worst-case ic-ratio.

Preliminary results (Appendix K) suggest that the DCT policy is more computationally efficient than the coloring-based policy on large, dense graphs, but is slightly worse in terms of performance. Further analysis of these results and possible improvements are left to future work. Our results, especially the residual decomposition of the VIS, provide a foundation for further on intervention design in more general settings.

## Funding transparency statement

Chandler Squires was supported by an NSF Graduate Research Fellowship and an MIT Presidential Fellowship and part of the work was performed during an internship at IBM Research. The work was supported by the MIT-IBM Watson AI Lab,

## Broader impact statement

Causality is an important concern in medicine, biology, econometrics and science in general (Pearl, 2009; Spirtes et al., 2000; Peters et al., 2017). A causal understanding of the world is required to correctly predict the effect of actions or external factors on a system, but also to develop fair algorithms. It is well-known that learning causal relations from observational data alone is not possible in general (except in special cases or under very strong assumptions); in these cases experimental ("interventional") data is necessary to resolve ambiguities.

In many real-world applications, interventions may be time-consuming or expensive, e.g. randomized controlled trials to develop a new drug or gene knockout experiments. These settings crucially rely on *experiment design*, or more precisely *intervention design*, i.e. finding a cost-optimal set of interventions that can fully identify a causal model. The ultimate goal of intervention design is accelerating scientific discovery by decreasing its costs, both in terms of actual costs of performing the experiments and in terms of automation of new discoveries.

Our work focuses on intervention design for learning causal DAGs, which have been notably employed as models in system biology, e.g. for gene regulatory networks (Friedman et al., 2000) or for protein signalling networks (Sachs et al., 2005). Protein signalling networks represent the way cells communicate with each other, and having reliable models of cell signalling is crucial to develop new treatments for many diseases, including cancer. Understanding how genes influence each other has also important healthcare applications, but is also crucial in other fields, e.g. agriculture or the food industry. Since even the genome of a simple organism as the common yeast contains 6275 genes, interventions like gene knockouts have to be carefully planned. Moreover, experimental design algorithms may prove to be a useful tool for driving down the time and cost of investigating the impact of cell type, drug exposure, and other factors on gene expression. These benefits suggest that there is a potential for experimental design algorithms such as ours to be a commonplace component of the future biological workflow.

In particular, our work establishes a number of new theoretical tools and results that 1) may drive development of new experimental design algorithms, 2) allow practitioners to estimate, prior to beginning experimentation, how costly their task may be, 3) offer an intervention policy that is able to run on much larger graphs than most of the related work, and provides more efficient intervention schedules than the rest.

Importantly, our work and in general intervention design algorithms have some limitations. In particular, as we have mentioned in the main paper, all these algorithms have relatively strong assumptions (e.g. no latent confounders or selection bias, infinite observational data, noiseless interventions, or in some case limitations on the graph structure (Greenewald et al., 2019)). If these assumptions are not satisfied in the data, or the practitioner does not realize their importance, the outcome of these algorithms could be misinterpreted or over-interpreted, leading to wasteful experiments or overconfident causal conclusions. Wrong causal conclusions may lead to potentially severe unintended side effects or unintended perpetuation of bias in algorithms.

Even in case of correct causal conclusions, the actualized impact of experimental design depends on the experiments in which it is used. Potential positive uses cases include decreasing the cost of drug development, in turn leading to better and cheaper medicine for consumers.

## Footnotes

[1]A code base to recreate these results can be found at https://github.com/csquires/dct-policy.

[2]A node is *dominated* if all incident edges are directed, or if it has only a single incident edge to a neighbor with more than one incident undirected edges

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
