[Supplementary Material · DCT_policy_supp.pdf]

## Supplementary material for: Active Structure Learning of Causal DAGs via Directed Clique Trees

## A    Meek Rules

In this section, we recall the *Meek rules* (Meek, 1995) for propagating orientations in DAGs. Of the standard four Meek rules, two of them only apply when the DAG contains v-structures. Since all DAGs that we need to consider do not have v-structures, we include only the first two rules here.

**Proposition 2** (Meek Rules under no v-structures)**.**

1. ***No colliders:*** *If $a \to_G b -_G c$ and $a$ is not adjacent to $c$, then $b \to_G c$.*

2. ***Acyclicity:*** *If $a \to_G b \to_G c$ and $a$ is adjacent to $c$, then $a \to_G c$.*

## B    The running intersection property

A useful and well-known property of clique trees, used throughout proofs in the remainder of the appendix, is the following:

**Prop.** (Running intersection property)**.** *Let $\gamma = \langle C_1, \ldots, C_K \rangle$ be the path between $C_1$ and $C_K$ in the clique tree $T_G$. Then $C_1 \cap C_K \subseteq C_k$ for all $C_k \in \gamma$.*

We refer the interested reader to Maathuis et al. (2018).

## C    Proof of Proposition 1

This proposition describes the connection between arrow-meets and intersection comparability. In order to prove this proposition, we begin by establishing the following propositions:

**Proposition 3.** *Suppose $C_1$ and $C_2$ are adjacent in $T_G$. Then for all $v_1 \in C_1 \setminus C_2$, $v_2 \in C_2 \setminus C_1$, $v_1$ and $v_2$ are not adjacent in $G$.*

*Proof.* We prove the contrapositive. Suppose $v_1 \in C_1 \setminus C_2$ and $v_2 \in C_2 \setminus C_1$ are adjacent. Then $C_3' = (C_1 \cap C_2) \cup \{v_1, v_2\}$ is a clique and belongs to some maximal clique $C_3$. For the induced subtree property to hold, $C_3$ must lie between $C_1$ and $C_2$, i.e., $C_1$ and $C_2$ are not adjacent. □

**Proposition 4.** *Let $D$ be a moral DAG, there are no undirected edges in any of its directed clique trees $T_D$, and therefore neither in its directed clique graph $\Gamma_D$.*

*Proof.* (By contradiction). Suppose $v_1 \to_D v_{12}$ for $v_1 \in C_1 \setminus C_2$ and $v_{12} \in C_1 \cap C_2$. Suppose $v_2 \to_D v'_{12}$ for $v_2 \in C_2 \setminus C_1$, and $v'_{12} \in C_1 \cap C_2$. By the assumption that $D$ does not have v-structures and by Prop. 3, $v_{12} \neq v'_{12}$. Similarly, since $v_{12} \to_D v_2$ (otherwise there would be a v-structure with $v_1 \to_D v_{12}$) and $v'_{12} \to_D v_1$ (otherwise there would be a collider with $v_2 \to_D v'_{12}$). However, this induces a cycle $v_1 \to_D v_{12} \to_D v_2 \to_D v'_{12} \to_D v_1$. □

Now we can finally prove the final proposition:

**Proposition 1.** *Suppose $C_1 \ast\!\!\to_{T_D} C_2$ and $C_2 \leftarrow\!\!\ast_{T_D} C_3$ in $T_D$. Then these edges are intersection comparable. Equivalently in the contrapositve, if $C_1 \ast\!\!\to_{T_D} C_2$ and $C_2 \ast\!\!-\!\ast_{T_D} C_4$ are intersection incomparable, we can immediately deduce that $C_2 \to_{T_D} C_4$.*

*Proof.* We prove the contrapositive. If $C_1 \cap C_2 \not\subseteq C_2 \cap C_3$ and $C_1 \cap C_2 \not\supseteq C_2 \cap C_3$, then there exist nodes $v_{12} \in (C_1 \cap C_2) \setminus C_3$ and $v_{23} \in (C_2 \cap C_3) \setminus C_1$. Since $v_{12}$ and $v_{23}$ are both in the same clique $C_2$ they are adjacent in the underlying DAG $D$, i.e. $v_{12} -_D v_{23}$. Moreover since $C_1 \ast\!\!\to_{T_D} C_2$ by the definition of a directed clique graph, this edge is oriented as $v_{12} \to_D v_{23}$. Then by Prop. 4, $C_2 \to_{T_D} C_3$. □

Figure 7: A DAG, its DCT with a conflicting source, and its DCG without a conflicting source.

## D   Proof of Lemma 2

**Lemma 2.** *For any moral DAG D, one can always construct a CDCT with no arrow-meets.*

*Proof.* To construct a CDCT with no arrow-meets, our approach is to first construct the DCT in a special way, so that after contraction, there are no arrow-meets. In particular, we need a DCT such that each bidirected component has at most one incoming edge. A DCT in which this does not hold is said to have *conflicting sources*, formally:

**Definition 11.** *A directed clique tree $T_D$ has two* conflicting sources $C_0$ and $C_{K+1}$, if $C_0 \to_{T_D} C_1$ *and $C_K \leftarrow_{T_D} C_{K+1}$, and $C_1$ and $C_K$ are part of the same bidirected component $B \in \mathcal{B}(T_D)$, i.e. $C_1, C_K \in B$, possibly with $C_1 = C_K$.*

An example of a clique tree with conflicting sources is given in Fig. 7. The first DCT has conflicting sources $\{1, 2\}$ and $\{2, 3, 4\}$, while the second DCT does not have conflicting sources.

We will now show that Algorithm 3 constructs a DCT with no conflicting sources. This is sufficient to prove Lemma 2, since after contraction, the resulting CDCT will have no arrow-meets.

First, Algorithm 3 constructs a weighted clique graph $W_G$, which is a complete graph over vertices $\mathcal{C}(G)$, with the edge $C_1 -_{W_G} C_2$ having weight $|C_1 \cap C_2|$. We will show that at each iteration $i$, there are no conflicting sources in $T_D$. This is clearly true for $i = 0$ since $T_D$ has no edges to begin.

At a given iteration $i$, suppose that the candidate edge $e = C_1 \ast\!\!\to C_2$ is a maximum-weight edge that does not create a cycle, i.e. $e \in E$, but that it will induce conflicting sources. That is, the current $T_D$ already contains $C_2 \leftarrow\!\ast C_3 \leftarrow\!\ast \ldots \leftarrow\!\ast C_{K-1} \leftarrow C_K$, where we choose $C_K$ that has no parents. Note that we can do this by following any directed/bidirected edges upstream (away from $C_2$), which must terminate since $T_D$ is a tree and thus does not have cycles.

By Prop. 1, $C_1 \cap C_2 \lesseqgtr C_2 \cap C_3$. In this case, $C_1 \cap C_2 \subseteq C_2 \cap C_3$, since $C_2 \leftarrow\!\ast C_3$ was already picked as an edge and thus cannot have less weight (in other words, it cannot have a smaller intersection) than $C_1 \ast\!\!\to C_2$. Furthermore, since $C_1 - C_2 - C_3$ is a valid subgraph of the clique tree, we must have $C_1 \cap C_3 \subseteq C_2$ by the running intersection property of clique trees (see Appendix B). Combined with $C_1 \cap C_2 \subseteq C_2 \cap C_3$, we have $C_1 \cap C_3 = C_1 \cap C_2$. This means that $C_1 - C_3$ is also a valid edge in the weighted clique graph and it has the same weight ($C_1 \cap C_3$) as the $C_1 - C_2$ edge ($C_1 \cap C_2$). Moreover since $C_1 \ast\!\!\to C_2$ then this edge will also preserve the same orientations $C_1 \ast\!\!\to C_3$. Thus, $C_1 \ast\!\!\to C_3$ is another candidate maximum-weight edge that does not create a cycle. We may continue this argument, replacing $C_2$ by $C_k$, to show that $C_1 \ast\!\!\to C_K$ is a maximum weight edge that does not create a cycle. Since $C_K$ has no parents, there are still no conflicting sources after adding $C_1 \ast\!\!\to C_K$. Since we always pick a maximum-weight edge that does not create a cycle, this algorithm creates a maximum-weight spanning tree of $W_G$ (Koller & Friedman, 2009), which is guaranteed to be a clique tree of $G$ Koller & Friedman (2009). $\qquad\square$

## E   Proof of Theorem 1

We restate the theorem here:

**Theorem 1.** *A single-node intervention set is a VIS for any general DAG D iff it contains VISes for each residual $R \in \mathcal{R}(\tilde{T}_G)$ for all chain components $G \in \mathcal{CC}(\mathcal{E}(D))$ of its essential graph $\mathcal{E}(D)$.*

---

**Algorithm 3** CONSTRUCT_DCT

1: **Input:** DAG $D$
2: let $W_G$ be the weighted clique graph of $G = \mathrm{skel}(D)$
3: let $T_D$ be the empty graph over $V(W_G)$
4: **for** $i = 1, \ldots, |V(W_G)| - 1$ **do**
5:     let $E$ be the set of maximum-weight edges of $W_G$ that do not create a cycle when added to $T_D$
6:     select $e \in E$ s.t. there are no conflicting sources
7:     add $e$ to $T_D$
8: **end for**
9: Contract the bidirected components of $T_D$ and create the CDCT $\tilde{T}_D$
10: **Return** $\tilde{T}_D$

---

Figure 8: A DAG, its contracted directed clique tree, its residuals, and its residual essential graph.

In order to prove the following theorem we start by introducing a few useful concepts and results.

### E.1 Residual essential graphs

The residuals decompose the DAG into parts which must be separately oriented. Intuitively, after adding orientations *between* all pairs of residuals, the inside of one residual is cut off from the insides of other residuals. The following definition and lemmas formalize this intuition.

**Definition 12.** *The* residual essential graph $\mathcal{E}_{res}(D)$ *of $D$ has the same skeleton as $D$, with $v_1 \to_{\mathcal{E}_{res}(D)} v_2$ iff $v_1 \to_D v_2$ and $v_1$ and $v_2$ are in different residuals of $\tilde{T}_D$.*

The following lemma establishes that after finding the orientations of edges in the DCT, the only remaining unoriented edges are in the residuals.

**Lemma 3.** *The oriented edges of $\mathcal{E}_{res}(D)$ can be inferred directly from the oriented edges of $T_D$.*

*Proof.* In order to prove this theorem, we first introduce an alternative characterization of the residual essential graph defined only in terms of the orientations in the contracted DCT and prove its equivalence to Definition 12. Let $\mathcal{E}'_{res}(D)$ have the same skeleton as $D$, with $i \to_{\mathcal{E}'_{res}(D)} j$ if and only if $j \in \mathrm{Res}_{\tilde{T}_D}(B)$ and $i \in P$, for some $B \in \mathcal{B}(T_D)$ and its unique parent $P$.

Suppose $v_1 \to_D v_2$ for $v_1 \in R_1$ and $v_2 \in R_2$, with $R_1, R_2 \in \mathcal{R}(\tilde{T}_D)$ and $R_1 \neq R_2$. Let $R_1 = \mathrm{Res}_{\tilde{T}_D}(B_1)$ and $R_2 = \mathrm{Res}_{\tilde{T}_D}(B_2)$ for $B_1, B_2 \in \mathcal{B}(\tilde{T}_D)$. There must be at least one clique $C_1 \in B_1$ that contains $v_1$, and likewise one clique $C_2 \in B_2$ that contains $v_2$. Since $v_1$ and $v_2$ are adjacent, by the induced subtree property there must be some maximal clique on the path between $C_1$ and $C_2$ which contains $v_1$ and $v_2$. Let $C_{12}$ be the clique on this path containing $v_1$ and $v_2$ that is closest to $C_1$. Then, the next closest clique to $C_1$ must not contain $v_2$, so we will call this clique $C_{1\setminus 2}$. Since $v_1 \to_D v_2$, we know that $C_{1\setminus 2} \to_{T_D} C_{12}$, hence $C_{1\setminus 2}$ and $C_{12}$ are in different bidirected components, and thus $v_1 \to_{\mathcal{E}_{res}(D)} v_2$. $\square$

**Lemma 4.** *The $\mathcal{E}_{res}(D)$ is complete under Meek's rules (Meek, 1995).*

*Proof.* Since Meek rules are sound and complete rules for orienting PDAGs (Meek, 1995), and in our setting only two of the Meek rules apply (see Prop. 2 in Appendix A), it suffices to show that neither applies for residual essential graphs.

First, suppose $i \to_{\mathcal{E}_{\text{res}}(D)} j$ and $j \to_{\mathcal{E}_{\text{res}}(D)} k$. We must show that if $i$ and $k$ are adjacent, then $i \to_{\mathcal{E}_{\text{res}}(D)} k$, i.e. the acyclicity Meek rule does not need to be invoked.

We use the alternative characterization of $\mathcal{E}_{\text{res}}(D)$ from the proof of Lemma 3, which establishes that $i \to_{\mathcal{E}} j$ iff. $j \in \text{Res}_{\mathcal{T}_D}(B)$ and $i \in P$ for some $B \in \mathcal{B}(T_D)$ and its unique parent $P$.

Since $j \to_{\mathcal{E}_{\text{res}}(D)} k$, there must exist some component $B_{jk} \in \mathcal{B}(T_D)$ containing $j$ and $k$ whose parent component $B_{j\backslash k}$ contains $j$ but not $k$, i.e. $B_{j\backslash k} \to_{\tilde{T}_D} B_{jk}$. Likewise, there must be a component $B_{ij}$ containing $i$ and $j$ whose parent component $B_{i\backslash j}$ contains $i$ but not $j$, i.e. $B_{i\backslash j} \to_{\tilde{T}_D} B_{ij}$. Moreover, since there is a clique on $\{i, j, k\}$, there must be at least one component $B_{ijk}$ containing $i$, $j$ and $k$.

We will prove that $B_{jk}$ and $B_{j\backslash k}$ both contain $i$, which implies $i \to_{\tilde{T}_D} k$.

Let $\gamma$ be the path in $\tilde{T}_D$ between $B_{i\backslash j}$ and $B_{jk}$. This path must contain the edge $B_{j\backslash k} \to B_{jk}$, since $B_{i\backslash j}$ is upstream of $B_{jk}$, and $\mathcal{T}_D$ is a tree. By the induced subtree property on $k$, no component on the path other than $B_{jk}$ can contain $k$. Now consider the path between $B_{ijk}$ and $B_{i\backslash j}$. By the induced subtree property on $k$, this path must pass through $B_{jk}$. Finally, by the induced subtree property on $i$, $B_{jk}$ and $B_{j\backslash k}$ must both contain $i$.

Now, we prove that also the first Meek rule is not invoked. Suppose $i \to_{\mathcal{E}_{\text{res}}(D)} j$, and $j$ is adjacent to $k$. We must show that if $i$ is not adjacent to $k$, then $j \to_{\mathcal{E}_{\text{res}}(D)} k$.

Since $\{i, j, k\}$ do not form a clique, there must be distinct components containing $i \to j$ and $j \to k$. Let $B_{ij}$ and $B_{jk}$ denote the closest such components in $\tilde{T}_D$, which are uniquely defined since $\tilde{T}_D$ is a tree. Since $i$ is upstream of $k$, $B_{ij}$ must be upstream of $B_{jk}$. Let $P := \text{pa}_{\tilde{T}_D}(B_{jk})$, we know $j \in P$ since it is on the path between $B_{ij}$ and $B_{jk}$ (it is possible that $P = B_{ij}$). Since we picked $B_{jk}$ to be the closest component to $B_{ij}$ containing $\{j, k\}$, we must have $k \notin P$, so indeed $j \to_G k$. $\qquad\square$

For an example of the residual essential graph, see Fig. 8. Lemma 4 implies that the residuals must be oriented separately, since the orientations in one do not impact the orientations in others.

### E.2   Proof for a moral DAG

We then prove the result for a moral DAG $D$:

**Lemma 5** (VIS Decomposition)**.** *An intervention set is a VIS for a moral DAG $D$ iff it contains VISes for each residual of $\tilde{T}_D$. This implies that finding a VIS for $D$ can be decomposed in several smaller tasks, in which we find a VIS for each of the residuals in $\mathcal{R}(\tilde{T}_D)$.*

*Proof.*
**VISes of residuals are necessary.** We first prove that any VIS $\mathcal{I}$ of $D$ must contain VISes for each residual of $D$. Consider the residual essential graph $\mathcal{E}_{\text{res}}(D)$ of $D$. We show that if we intervene on a node $c_1$ in the residual $R_1 = \text{Res}_{\tilde{T}_D}(B_1)$ of some $B_1 \in \mathcal{B}(\tilde{T}_D)$, then the only new orientations are between nodes in $R_1$, or in other words, each residual needs to be oriented independently.

By Definition 12, all edges between nodes in different residuals are already oriented in $\mathcal{E}_{\text{res}}(D)$. A new orientation between nodes in $R_1$ will not have any impact for the nodes in the other residuals, which we can show by proving that Meek rules described in Prop. 2 would not apply outside of the residual. In particular, Meek Rule 1 does not apply at all, since $b$ and $c$ must be in the same residual since the edge is undirected, but then $a$ is adjacent to $c$ since it's a clique. Likewise, $a -_{\mathcal{E}_{\text{res}}(d)} c$, then $a$ and $b$ are in the same residual, so Meek Rule 2 only orients edges with both endpoints in the same residual.

**VISes of residuals are sufficient.** Now, we show that if $\mathcal{I}$ contains VISes for each residual of $D$, then it is a VIS for $D$, i.e. that orienting the residuals will orient the whole graph by applying recursively Meek rules. We will accomplish this by inductively showing that all edges in each bidirected component are oriented. Let $\gamma = \langle B_1, \ldots, B_n \rangle$ be a path from the root of $\tilde{T}_D$ to a leaf of $\tilde{T}_D$. As our base case, all edges in $B_1$ are oriented, since $B_1 = \text{Res}_{\tilde{T}_D}(B_1)$. Now, as our induction hypothesis, suppose that all edges in $B_{i-1}$ are oriented.

The edges between nodes in $B_i$ are partitioned into three categories: edges with both endpoints also in $B_{i-1}$, edges with both endpoints in $\text{Res}_{\tilde{T}_D}(B_i)$, and edges with one endpoint in $B_{i-1}$ and one

---

**Algorithm 4** FIND_MVIS_DCT

---

 1: **Input:** Moral DAG $D$
 2: let $\tilde{T}_D$ be the contracted directed clique tree of $D$
 3: let $S = \emptyset$
 4: **for** component $B$ of $T_D$ **do**
 5:     let $R = \mathrm{Res}_{\tilde{T}_D}(B)$
 6:     let $S' = $ FIND_MVIS_ENUMERATION$(G[R])$
 7:     let $S = S \cup S'$
 8: **end for**
 9: **Return** $S$

---

---

**Algorithm 5** FIND_MVIS_ENUMERATION

---

 1: **Input:** DAG $D$
 2: **if** $D$ is a clique **then**
 3:     Let $\pi$ be a topological ordering of $D$
 4:     Let $S$ include even-indexed element of $\pi$
 5:     **Return** $S$
 6: **end if**
 7: **for** $s = 1, \ldots, |V(D)|$ **do**
 8:     **for** $S \subseteq V(D)$ with $|S| = s$ **do**
 9:         **if** $S$ fully orients $D$ **then**
10:             **Return** $S$
11:         **end if**
12:     **end for**
13: **end for**

---

endpoint in $\mathrm{Res}_{\tilde{T}_D}(B_i)$. The first category of edges are directed by the induction hypothesis, and the second category of edges are directed by the assumption that $\mathcal{I}$ contains VISes for each residual. It remains to show that all edges in the third category are oriented. Each of these edges has one endpoint in some $C_{i-1} \in B_{i-1}$ and one endpoint in some $C_i$ in $B_i$, so we can fix some $C_{i-1}$ and $C_i$ and argue that all edges from $C_{i-1} \cap C_i$ to $C_i \setminus C_{i-1}$ are oriented.

Since $C_{i-1} \to_{R_D} C_i$, there exists some $c_{i-1} \in C_{i-1} \setminus C_i$ and $c' \in C_i \cap C_{i-1}$ such that $c_{i-1} \to_D c'$. By Prop. 3, $c_{i-1}$ is not adjacent to any $c_i \in C_i \setminus C_{i-1}$, so Meek Rule 1 ensures that $c' \to_D c_i$ is oriented. For any other node $c'' \in C_{i-1} \cap C_i$, either $c' \to_D c''$, in which case Meek Rule 2 ensures that $c_{i-1} \to_D c''$ and the same argument applies, or $c'' \to_D c'$, in which case Meek Rule 2 ensures that $c'' \to_D c_i$. $\qquad\square$

### E.3 Proof for a general DAG

We can now easily prove the theorem for any DAG $D$:

**Theorem 1.** *A single-node intervention set is a VIS for any general DAG $D$ iff it contains VISes for each residual $R \in \mathcal{R}(\tilde{T}_G)$ for all chain components $G \in \mathcal{CC}(\mathcal{E}(D))$ of its essential graph $\mathcal{E}(D)$.*

*Proof.* By the previous result (Lemma 5) and Lemma 1 from (Hauser & Bühlmann, 2014). $\qquad\square$

## F Algorithm for finding an MVIS

An algorithm using the decomposition into residuals to compute a minimal verifying intervention set (MVIS) is described in Algorithms 4 and 5. Compared to running Algorithm 5 on any moral DAG, using Algorithm 4 ensures that we only have to enumerate over subsets of the nodes in each residual, which in general require far fewer interventions. Moreover, the residual of any component containing a single clique is itself a clique, which have easily characterized MVISes, and Algorithm 5 efficiently computes.

# G Proof of Theorem 2

First, we prove the following proposition:

**Proposition 5.** *Let $D$ be a moral DAG, $\mathcal{E} = \mathcal{E}(D)$ and let $\tilde{T}_D$ contain a single bidirected component. Then $m(D) \geq \left\lfloor \frac{\omega(\mathcal{E})}{2} \right\rfloor$.*

*Proof.* Let $C_1 \in \arg\max_{C \in \mathcal{C}(\mathcal{E})} |C|$. By the running intersection property (see Appendix B), for any clique $C_2$, $C_1 \cap C_2 \subseteq C_2 \cap C_{\text{adj}}$ for $C_{\text{adj}}$ adjacent to $C_2$ in $T_D$. Since $C_{\text{adj}} \leftrightarrow_{T_D} C_2$, we have $v_{12} \to_D v_{2\backslash 1}$ for all $v_{12} \in C_1 \cap C_2$ and $v_{2\backslash 1} \in C_2 \setminus C_1$, i.e. there is no node in $D$ outside of $C_1$ that points into $C_1$. Thus, since the Meek rules only propagate downward, intervening on any nodes outside of $C_1$ does not orient any edges within $C_1$. Finally, since $C_1$ is a clique, each consecutive pair of nodes in the topological order of $C_1$ must have at least one of the nodes intervened in order to establish the orientation of the edge between them. This requires at least $\left\lfloor \frac{|C_1|}{2} \right\rfloor$ interventions, achieved by intervening on the even-numbered nodes in the topological ordering. $\qquad\square$

Now we can prove the following result for a moral DAG $D$:

**Lemma 6.** *Let $D$ be a moral DAG and let $G = \text{skel}(D)$. Then $m(D) \geq \left\lfloor \frac{\omega(G)}{2} \right\rfloor$, where $\omega(G)$ is the size of the largest clique in $G$.*

Consider a path $\gamma$ from the source of $\tilde{T}_D$ to the bidirected component containing the largest clique, i.e., $\gamma = \langle B_1, \ldots, B_Z \rangle$. For each component, pick $C_i^* \in \arg\max_{C \in B_i} |C|$. Also, let $R_i = \text{Res}_{\tilde{T}_D}(B_i)$. We will prove by induction that $\sum_{i=1}^{z} m(D[R_i]) \geq \max_{i=1}^{z} \left\lfloor \frac{|C_i^*|}{2} \right\rfloor$ for any $z = 1, \ldots, Z$. As a base case, it is true for $z = 1$, since $R_1 = B_1$ and by Prop. 5.

Suppose the lower bound holds for $z - 1$. If $C_z^*$ is not the unique maximizer of $\left\lfloor \frac{|C_z^*|}{2} \right\rfloor$ over $i = 1, \ldots, z$, the lower bound already holds. Thus, we consider only the case where $B_z$ is the unique maximizer.

Let $S_z = C_z^* \cap B_{z-1}$. By the running intersection property (see Appendix B), $S_z$ is contained in the clique $C_{\text{adj}}$ in $B_{z-1}$ which is adjacent to $C_z^*$ in $T_D$. Since $C_{\text{adj}}$ is distinct from $C_z^*$, $|C_{\text{adj}}^*| \geq |S_z| + 1$, and by the induction hypothesis we have that

$$
\begin{aligned}
\sum_{i=1}^{z-1} m(D[R_i]) &\geq \max_{i=1,\ldots,z-1} \left\lfloor \frac{|C_i^*|}{2} \right\rfloor \\
&\geq \left\lfloor \frac{|C_{z-1}^*|}{2} \right\rfloor \\
&\geq \left\lfloor \frac{|C_{\text{adj}}|}{2} \right\rfloor \\
&\geq \left\lfloor \frac{|S_z| + 1}{2} \right\rfloor
\end{aligned}
$$

Finally, applying Prop. 5,

$$
\begin{aligned}
\left\lfloor \frac{|S_z + 1|}{2} \right\rfloor + m(D[R_z]) &\geq \left\lfloor \frac{|S_z| + 1}{2} \right\rfloor + \left\lfloor \frac{|C_z^* \cap R_z|}{2} \right\rfloor \\
&\geq \left\lfloor \frac{|C_z^*|}{2} \right\rfloor
\end{aligned}
$$

where the last equality holds since $|S_z| + |C_z^* \cap R_z| = |C_z^*|$ and by the property of the floor function that $\left\lfloor \frac{a+1}{2} \right\rfloor + \left\lfloor \frac{b}{2} \right\rfloor \geq \left\lfloor \frac{a+b}{2} \right\rfloor$, which can be easily checked.

Finally we can prove the theorem:

---

**Algorithm 6** CLIQUEINTERVENTION

---

1: **Input:** Clique $C$
2: **while** $C -_{\Gamma_D} C'$ unoriented for some $C'$ **do**
3:    **if** $\exists v$ non-dominated in $C$ **then**
4:       Pick $v \in C$ at random among non-dominated nodes.
5:    **else**
6:       Pick $v \in C$ at random.
7:    **end if**
8:    Intervene on $v$.
9: **end while**
10: **Output:** $P_{\text{up}}(C)$

---

---

**Algorithm 7** EDGEINTERVENTION

---

1: **Input:** Adjacent cliques $C, C'$
2: **while** $C -_{\Gamma_D} C'$ unoriented **do**
3:    Pick $v \in C \cap C'$ at random.
4:    Intervene on $v$.
5: **end while**
6: **Output:** $P_{\text{up}}(C)$

---

**Theorem 2.** *Let $D$ be any DAG. Then $m(D) \geq \sum_{G \in cc(\mathcal{E}(D))} \left\lfloor \frac{\omega(G)}{2} \right\rfloor$, where $\omega(G)$ is the size of the largest clique in each of the chain components $G$ of the essential graph $\mathcal{E}(D)$.*

*Proof.* By Lemma 6 and Lemma 1 in Hauser & Bühlmann (2014). $\qquad\qquad\square$

## H  Clique and Edge Interventions

We present the procedures that we use for clique- and edge-interventions in Algorithm 6 and Algorithm 7, respectively.

## I  Identify-Upstream Algorithm

Given the clique graph, a simple algorithm to identify the upstream branch consists of performing an edge-intervention on each pair of parents of $C$ to discover which is the most upstream. However, if the number of parents of $C$ is large, this may consist of many interventions. The following lemma establishes that the only parents which are candidates for being the most upstream are those whose intersection with $C$ is the smallest:

**Proposition 6.** *Let $P_{up}(C) \in pa_{\Gamma_D}(C)$ be the parent of $C$ which is upstream of all other parents. Then $P_{up}(C) \in \mathcal{P}_{\Gamma_D}(C)$, where $\mathcal{P}_{\Gamma_D}(C)$ is the set of parents of $C$ in $\Gamma_D$ with the smallest intersection size, i.e., $P \in \mathcal{P}_{\Gamma_D}(C)$ if and only if $P \to_{\Gamma_D} C$ and $|P \cap C| \leq |P' \cap C|$ for all $P' \in pa_{\Gamma_D}(C)$.*

*Proof.* We begin by citing a useful result on the relationship between clique trees and clique graphs when the clique contains an intersection-comparable edge:

**Lemma 7** (Galinier et al. (1995))**.** *If $C_1 -_{T_G} C_2 -_{T_G} C_3$ and $C_1 \cap C_2 \subseteq C_2 \cap C_3$, then $C_1 -_{\Gamma_G} C_3$.*

**Corollary 1.** *If $C_1 -_{T_G} C_2 -_{T_G} C_3$ and $C_1 \cap C_2 \subseteq C_2 \cap C_3$, then $C_1 \cap C_3 = C_1 \cap C_2$.*

*Proof.* By the running intersection property of clique trees (see Appendix B), $C_1 \cap C_3 \subseteq C_2$. Combined with $C_1 \cap C_2 \subseteq C_2 \cap C_3$ and simple set logic, the result is obtained. $\qquad\square$

Every parent of $C$ is adjacent in $\Gamma_D$ to every other parent of $C$ by Prop. 1 and Lemma 7, and since every edge has at least one arrowhead, there can be at most one parent of $C$ that does not have an incident arrowhead.

**Algorithm 8** IDENTIFYUPSTREAM

1: **Input:** Clique $C$
2: **for** $P_1, P_2 \in \mathcal{P}_{\Gamma_D}(C)$ **do**
3:    perform an edge-intervention on $P_1 -_{\Gamma_D} P_2$
4: **end for**
5: **Output:** $P_{\text{up}}(C)$

---

Now we show that this parent must be in $\mathcal{P}_{\Gamma_D}(C)$. Corollary 1 implies that for any triangle in $\Gamma_G$, two of the edge labels (corresponding to intersections of their endpoints) must be equal. If $P \in \mathcal{P}_{\Gamma_D}(C)$ and $P' \in \text{pa}_{T_D}(C) \setminus \mathcal{P}_{\Gamma_D}(C)$, then the labels of $P \rightarrow_{\Gamma_D} C$ and $P' \rightarrow_{\Gamma_D} C$ are of different size and thus cannot match. Therefore, the label of $P \cap P' = P \cap C$. Finally, since we already know $P \rightarrow_{\Gamma_D} C$, it must also be the case that $P \rightarrow_{\Gamma_D} P'$. □

## J  Proof of Theorem 3

We start by proving bounds for each of the two phases:

**Lemma 8.** *Algorithm 2 uses at most $\lceil \log_2 |\mathcal{C}| \rceil$ clique-interventions. Moreover, assuming $T_G$ is intersection-incomparable, Algorithm 2 uses no edge-interventions.*

*Proof.* Since $T_G$ is intersection-incomparable, after a clique-intervention on $C$, orientations propagate in all but at most one branch of $T_G$ out of $C$. By the definition of a central node, the one possible remaining branch has at most half of the nodes from the previous time step, so the number of edges in $T_G$ reduces by at least half after each clique-intervention. Thus, there can be at most $\lceil \log_2 |\mathcal{C}| \rceil$ clique-interventions. □

For ease of notation, we will overload the symbol CC for the chain components of a chain graph $G$ to take a DAG as an argument, and return the subgraphs corresponding to the chain components of its essential graph. Formally, $\text{CC}(D) = \{D[V(G)] \mid G \in \text{CC}(\mathcal{E}(D))\}$.

**Lemma 9.** *The second phase of Algorithm 1 (line 6-8) uses at most $\sum_{C \in \mathcal{C}(D')} |\text{Res}_{\tilde{T}_{D'}}(C)| - 1$ single-node interventions for the moral DAG $D' \in \mathcal{CC}(D)$.*

*Proof.* Eberhardt et al. (2006) show that $n - 1$ single-node interventions suffice to determine the orientations of all edges between $n$ nodes. We sum this value over all residuals. □

**Theorem 3.** *Assuming $\Gamma_G$ is intersection-incomparable, Algorithm 1 uses at most $(3\lceil \log_2 \mathcal{C}_{max} \rceil + 2)m(D)$ single-node interventions, where $\mathcal{C}_{max} = \max_{G \in \mathcal{CC}(\mathcal{E}(D))} |\mathcal{C}(G)|$.*

*Proof.* Consider a moral DAG $D' \in \text{CC}(D)$. We will show that Algorithm 1 uses at most $(3\lceil \log_2 |\mathcal{C}(\mathcal{E}(D))| \rceil + 2)m(D')$ single-node interventions. The result then follows since $m(D) = \sum_{D' \in \text{CC}(D)} m(D')$, the total number of interventions used by Algorithm 1 is the sum over the number interventions used for each chain component, and $\mathcal{C}_{max} \geq |\mathcal{C}(\mathcal{E}(D))|$ for all $D'$.

Assume that for each clique-intervention in Algorithm 2, we intervene on every node in the clique. Then, the number of single-node interventions used by each clique intervention is upper-bounded by $\omega(G)$. By Theorem 2 and the simple algebraic fact that $\forall a \in \mathbb{N}, a \leq 3\lfloor \frac{a}{2} \rfloor$ (which can be proven simply by noting that if $a$ is even $a \leq 3\frac{a}{2}$ and if $a$ is odd $a \leq 3\frac{a-1}{2}$., $\omega(G) \leq 3m(D)$, Algorithm 2 uses at most $3m(D)$ single-node interventions. Next, by Lemma 5 and Lemma 9, and the fact that $\forall a \in \mathbb{N}, a - 1 \leq 2\lfloor \frac{a}{2} \rfloor$, the second phase of Algorithm 1 uses at most $2m(D)$ single-interventions. □

## K  Additional Experimental Results

### K.1  Scalability of OptSingle

We use the same graph generation procedure as outlined in Section 5. We compare OptSingle, Coloring, DCT, and ND-Random on graphs of up to 25 nodes in Fig. 9. We observe that at 25 nodes,

(a) Average ic-ratio

(b) Average Computation Time

Figure 9: Comparison (over 100 random synthetic DAGs)

(a) Average Computation Time

`OptSingle` already takes more than 2 orders of magnitude longer than either the `Coloring` or DCT policies to select its interventions, while achieving comparable performance in terms of average competitive ratio.

### K.2 Computation time for large tree-like graphs

In this section, we report the results on average computation time associated with Fig. 6c from Section 5. We find similar scaling for our DCT policy and the `Coloring` policy, both taking about 5-10 seconds for graphs of up to 500 nodes, as seen in Fig. 10a.

### K.3 Comparison on large dense graphs

In this section, we generate dense graphs via the same Erdös-Rényi-based procedure described in Section 5. We show in Fig. 11 that the DCTpolicy is more scalable to dense graphs than the `Coloring` policy, but that our performance becomes slightly *worse* than even `ND-Random`. Since the size of the MVIS is already large for such dense graphs, this suggests that the two-phase nature of the DCTpolicy may be too restrictive for such a setting. Further analysis of the graphs on which different policies do well is left to future work.

(a) Average ic-ratio

(b) Average Computation Time

Figure 11: Comparison (over 100 random synthetic DAGs)