[Reviews · NeurIPS 2020]

Review 1

Summary and Contributions: -considers intervention design for orient essential graph into a DAG -active: the design is not fixed at once but sequentially taking into account -proves that the number of interventions necessary is at least the sum of half of the size of the largest cliques in each chain components - presents an algorithm for intervention design -which matches the bound up to a factor -which scales up higher and shows better worst case performance

Strengths: -mix of novel theoretical results and readily usable algorithm -quite clear simulations -new theory for orientation of DAGs from interventions

Weaknesses: -considering DAGs without latent confounders or cycles in the causal context is limiting -also only single node interventions are considered -simulation results do not show much improved average case performance

Correctness: The claims seem correct.

Clarity: The writing of the paper is relatively good, despite the easily fixable issues in additional feedback. Examples on the theoretical graph concepts are given as figures.

Relation to Prior Work: Relation to previous is described well in the paper: -extends central node algorithm of Greenewald NIPS 2019 that is for limited graph types to general DAGs -present all cases theoretical results in contrast to previous worst case results

Reproducibility: Yes

Additional Feedback: Minor writing issues: Terms noiseless and noisy interventions (Sec 1) are used before they are defined (Sec 2). Directed cycle is defined as sequence of edges with at least on directed edge, this is non-standard terminology that should be changes to something else. It is also troublesome since it does not correspond to acyclicity, one of the main assumptions of the paper. Intervention policy definition is unclear: a map rom EG to interventions? Is this now EGs for I-EGs? If it is just a map with EG as input, how can previous interventions affect it? Def 2 Is unclear and informal: 1) it seems to produce * which do not exist in Fig 1, 2) Fig 2 includes bidirected edges. Better to formulate in another way. Intuition on ic-ratio is missing, this should be added. Hard to interpret it as a reader. When is the result optimal? ----- AR acknowledged. This is good work but due to the clear limitations a higher grade is not warranted.


Review 2

Summary and Contributions: The paper studies intervention design for learning causal relationships between observed random variables. In brief, the problem setting is such that we are given a mixed graph G on node set V representing the Markov equivalence class of the observed distribution, and we want to design a sequence of interventions that allow us to infer the (causal) directions of the undirected edges (which are guaranteed to form a DAG); an intervention is an operation on node subset I that, from a purely combinatorial perspective, gives us the directions of the boundary edges of I. The task is then to find a shortest sequence of interventions that allows us to infer all edge directions. The paper focuses on the case of single-node adaptive intervention strategies, i.e. all interventions operate on single node at a time, and the selected interventions can depend on the results of prior interventions. The main results are (a) a lower bound for the number of single-node interventions required to fully orient a graph, and (b) an algorithm for generating a sequence of interventions with guarantees on the upper bound of interventions assuming structural conditions on the underlying true graph. The algorithm is implemented and compared against various prior proposals.

Strengths: Generally, this paper is quite far outside my own area, so I can't fully evaluate its significance in the context of causal inference community. The experimental results indicate that the proposed algorithm is somewhat better than the best prior proposal, and the lower bound results is reasonable.

Weaknesses: The results seem fairly incremental compared to prior work, and it's not clear if the results are interesting outside a small sub-community.

Correctness: The results seem be reasonable likely to be correct, however, due to the dense and non-self-contained presentation, I was not able to fully verify the details of the proofs within the review window. There seem to be various minor mistakes in the proofs, so thorough checking might be warranted. In the experimental section, a more direct comparison between DCT and Coloring would be appropriate, since they have at least on average very similar behaviour in terms of competitive ratio.

Clarity: While this paper is certainly outside my area, I found it to be not well written, and I had to frequently refer to prior work to make sense of the paper. On general level, the paper does not provide a good precise overview of the results, or provide context in terms of what theoretical bounds are obtained in prior work and how they compare to the present work. Some key details about the setting are omitted or mentioned only in passing. For example, the fact that all of the results consider only single-node interventions is mentioned only in a single sentence in the introduction, and is not reflected e.g. in Definition 7, which seems to define m(D) in terms of unbounded interventions. Likewise, the paper does not explain what information about edge directions an intervention I provides. The theoretical results are presented in a rather dense, front-loaded style, which requires the reader to keep track of lots of notation and abbreviations. However, the actual proofs don't seem very deep, and the presentation could probably be significantly streamlined.

Relation to Prior Work: Relevant related work seems to be cited sufficiently, but not sufficiently discussed to provide context for the results; in particular, the paper does not discuss what (if any) upper and lower bounds are provided in the prior work in a way that could be compared with the results in the present paper.

Reproducibility: Yes

Additional Feedback: - Line 42: "MEC" used before defined - Line 63: Definition of directed cycle looks weird, possibly should be *-> instead of *-*? (By this definition, e.g. 234 is a directed cycle in Fig1(a)) - Line 70: Parent set notation is not defined. - Section 5: How is m(D) computed in the experiments? I.e. is it the actual m(D), or the lower bound provided by Theorem 2? - Appendix, lines 591-593: Please elaborate on the clique intervention lower bound, or provide a reference. - Appendix, lines 613: The cited reference does not seem to have Lemma 1. Should be 2014 instead? Please include full details. -- update after rebuttal -- After the rebuttal and discussion, I'm slightly adjusting my score upwards. The lower bound is indeed kind of nice, but I still disagree with the authors on the clarity of presentation. The claim itself can be presented as a simple combinatorial statement, and the proof does not use any advanced techniques. In particular, I would encourage the authors to make sure that the proofs in the main paper can be followed without reference to the appendix or prior work. In the experiments, it looks like COLOURING and DCT both have very similar performance, and DCT does not (substantially) improve over COLOURING in practice (though DCT has theoretical guarantees in the restricted case). This is where I would have wanted to see the "more direct comparison" on instance-by-instance basis; since the average ic-ratio is generally slightly better for COLOURING, but maximum ic-ratio is worse. Is the bad maximum ic-ratio due to single outliers? Does COLOURING sometimes also produce much better ic-ratios than DCT to counterbalance the bad outliers? (I.e., I would want to see the ic-ratios of COLOURING and ICT plotted against each other for all instances in a certain size category.)


Review 3

Summary and Contributions: The authors considered the problem of experiment design to fully identify the causal structure by performing minimum number of interventions. They showed that for full identification of the causal graph, it is required to orient some specific induced subgraphs of the causal graph, which they called "residuals". Based on this observation, they provided a lower bound on the number of interventions necessary for full identification. Moreover, a two-phase intervention policy was proposed where in the first phase, Central Node algorithm (Greenewald et al., 2019) have been utilized to reduce the problem to identify orientations within the residuals. In the second phase, the algorithm orients the edges in each residual.

Strengths: - The authors proposed a universal lower bound on the number of interventions necessary for full identification of the causal graph. In particular, they showed that this bound is equal to the sum of half the size of the largest cliques in each chain component of the essential graph. Comparing with the result in previous work, this bound holds true for any DAG in the MEC which is an interesting result. - Under some assumptions on the causal graph, the two-phase intervention policy uses at most O(\log(|C_max|) where C_max is the highest number of maximal cliques in any chain component.

Weaknesses: ===After rebuttal=== I read the reviews and the rebuttal. I think the "intersection-incomparability" is a restrcitive assumption in analyzing the proposed algorithm. Although the paper has a nice result for universal lower bound on the number of interventions necessary for full identification of the causal graph. In overall, I decided to keep my score unchanged. ========= - I think it is required to discuss in which causal graphs, the "intersection-incomparability of the clique graph" holds true. Is it a restrictive assumption? - About C_max: It seems that the definition of C_max in line 54 is different from the one in Theorem 3. It would be great to clarify this issue. Moreover, the value of C_max can be in the order of graph size. I am not sure whether there exists other related work with the approximation factor of log(n). - In the experiments, the structure of large graphs is close to trees. It would be great if the authors can report running times of the proposed algorithm in structures other than trees.

Correctness: It seems that the claims and the proposed algorithm are correct. However, I did not go in details of the proofs.

Clarity: The paper is generally well written.

Relation to Prior Work: The authors mentioned previous work in Section 6.

Reproducibility: Yes

Additional Feedback:

[Author Response · NeurIPS 2020]

We thank the reviewers for their valuable feedback. We are pleased that they recognize the novelty of our theoretical results (R1), find the universality of the lower bound interesting (R3), and believe that prior work is described well (R1).

**Clarity (R2)**: Thank you for pointing out the need for more emphasis on our setting, such as the consideration of only single node interventions. We will emphasize this in the abstract and more clearly throughout the paper.

We will extend our comparison to previous bounds and emphasize our bound has a unique flavor, since it is *instance-wise* rather than minimax or an average over a MEC. As mentioned on line 292, Hauser and Buhlmann (2014) and Shanmugam et al. (2015) prove minimax bounds, which both involve the size of largest maximal clique. For single-node interventions, the bound of Shanmugam et al. (2015) specializes to exactly the size of the largest maximal clique - a trivial result, since the MEC contains a DAG with the largest maximal clique upstream of all other nodes. Meanwhile, our bound is much stronger: regardless of the position of the largest maximal clique, it is a fundamental barrier to structure learning; moreover, the bound over *all* cases is only a factor of two smaller than the bound on the *worst* case.

R2 points out the paper might appear dense and not accessible to non-expert audiences. We have strived to make the paper as clear and self-contained as possible, e.g., by providing several examples and an overview of all required concepts, including standard graphical models and graph theory definitions. On the other hand, we do appreciate the recommendations that will make it even more accessible and streamlined. For example, we will clarify which edges are learned by a single-node intervention: these are the edges incident on the intervened node, as well as any logical implications of these edges (via the application of Meek's rules, reviewed in Appendix A). Regarding the dense notation, in the main paper we have tried to introduce only notation and definitions necessary to state the main results and give an intuition of the proofs. In fact, any apparent simplicity in the proofs is partially the result of careful definitions.

**Novelty/Applicability (R1, R2)**: We disagree with the claims that this work is incremental or narrowly interesting. As pointed out by R1, this work contains several novel theoretical results. Most importantly, our characterization of verifying intervention sets is a clear step toward a better understanding of identification of causal structures, which has numerous applications. Many of our results are potentially useful in more realistic settings, e.g. with larger interventions. Furthermore, our results, especially the residual characterization, may serve as a template for similar results in the presence of latent variables or cycles.

**Experiments (R2, R3)**: As suggested, we will report the running time of the policies on large graphs that are not tree-like. As a preliminary result, we find that for 80-node graphs generated in the same manner as our smaller graphs, DCT is roughly 30% faster than Coloring. However, both policies take more than an order of magnitude longer on these denser graphs. The comparison between DCT and Coloring in Fig. 6c,d is direct, we ask R2 for further clarification.

**(R1.1) Directed cycle definition**: Thank you for pointing this out; the definition of a directed cycle should not include asterisks on the left-hand sides of each edge. The new definition will correspond to the standard one for chain graphs.

**(R1.2) Intervention policy definition**: We will clarify that a policy is a map from *interventional* essential graphs.

**(R1.3) Definition 2**: See Line 62: * is a wildcard; we will recall this usage in the definition to bolster clarity.

**(R1.4) ic-ratio**: The ic-ratio of a policy on a DAG measures how many interventions the policy uses to orient that DAG, relative to the minimum number of interventions that can orient it. A policy is optimal on an instance if the ic-ratio is 1.

**(R2.1) $m(D)$ computation**: We compute the actual $m(D)$ in our experiments. An efficient method is described in Algorithm 4 of the Appendix; we will add a reference to the algorithm in Section 5.

**(R2.2) Appendix 591-593**: We will add an explanation. Briefly, in a clique, each pair of nodes must have at least one member intervened to establish the orientation of the edge between them. The smallest set satisfying this criterion has size $\lfloor \frac{n}{2} \rfloor$. This is formally proven in Shanmugam et al. (2015), Theorem 4, where $n$ is assumed even for simplicity.

**(R2.3) Appendix 613**: Thank you for pointing this out; the reference should be 2014.

**(R3.2) $C_{\mathbf{max}}$ definition**: Thanks for the correction. $C_{\mathbf{max}}$ should be a number, as in Thm 3, not a set. $C_{\mathbf{max}}$ is maximum number of maximal cliques in any chain component of the essential graph. As mentioned in lines 228-231, $\log n$ scaling is optimal for trees (Greenewald et al., 2019). On the other extreme, if the essential graph is a single clique, the log factor disappears. To the best of our knowledge, no work has established approximation factors before.

**(R3.2) intersection-incomparability**: We will add a discussion of this condition. First, intersection-incomparable chordal graphs have been introduced under the name "uniquely representable chordal graphs" (Kumar and Madhavan 02), which we will mention more clearly, and include familiar classes of graphs such as proper interval graphs. Second, while the assumption is necessary for our analysis of the DCT policy, the policy still performs well on graphs that do not satisfy it, as demonstrated by our experiments, suggesting that the assumption is not too restrictive in practice.

[Meta-Review · NeurIPS 2020]

summary: The authors study the problem of identifying a causal DAG from observational and active (ide closed-loop) interventional data. They lower bound the minimal number of interventions required to orient any graph in terms of the size of the largest cliques in the essential graph of the causal model. pro: - minimal-cost, active causal identification is an important topic - new lower bound on complexity of active identification of a causal model - readily applicable identification algorithm that comes close to achieving the lower bound on a restricted set of causal models cons: - somewhat restricted setting of no unobserved confounders, perfect atomic interventions only - empirical results are somewhat weak meta review: Solid paper on an important topic with novel theoretical results and a practical algorithmic approach.